# PROBE INTO MULTI-AGENT ADVERSARIAL REINFORCEMENT LEARNING THROUGH MEAN-FIELD OPTIMAL CONTROL

## ABSTRACT

Multi-agent adversarial reinforcement learning (MaARL) has shown promise in solving adversarial games. However, the theoretical tools for MaARL's analysis is still elusive. In this paper, we take the first step to theoretically understand MaARL through mean-field optimal control. Specifically, we model MaARL as a mean-field quantitative differential game between two dynamical systems with implicit terminal constraints. Based on the game, we respectively study the optimal solution and the generalization of the fore-mentioned game. We first establish a two-sided extremism principle (TSEP) as a necessary condition for the optimal solution of the game. We then show that this TSEP is also sufficient given that the terminal time is sufficiently small. Based on the TSEP, a generalization bound for MaARL is further proposed. This bound does not explicitly rely on the dimensions, norms, or other capacity measures of the model, which are usually prohibitively large in deep learning. To our best knowledge, this is the first work on the theory of MaARL.

## 1 INTRODUCTION

Reinforcement learning (RL) (Sutton, 1988; Sutton and Barto, 1998; Barto et al., 1991), aimed at single-agent environments, has been successfully deployed in many application areas, including ethology (Dayan and Daw, 2008), economics (Jasmin et al., 2011), psychology (Leibo et al., 2018), and system control (Arel et al., 2010). It studies how artificial systems learn to predict the optimal action according to the current state. The agent in the system determines the best course of action to maximize its rewards, which also moves its state to the next state.

However, real-world RL agents inhabit natural environments also populated by other agents. Agents in these environments can interact with each other and modify each other's rewards via their actions. Based on this point, multi-agent adversarial reinforcement learning (MaARL) is proposed by Uther and Veloso (2003), in which adversarial neural networks are employed for solving games in adversarial environments (Mandlekar et al., 2017). MaARL is well suited for multi-party game problems such as autonomous driving (Behzadan and Munir, 2019; Pan et al., 2019), AI gaming (Mandlekar et al., 2017; Pinto et al., 2017; Zhang et al., 2020), and auction games (Bichler et al., 2021). Moreover, adversarial neural networks can also improve feature robustness and sample efficiency (Ma et al., 2018).

Despite the empirical popularity of MaARL, its theoretical understanding remains blank. We attribute such a gap between theory and practice to the lack of analysis tool: the optimization objective is defined by a dynamical system, which is too complex to analyze directly.

In this paper, we aim to provide new theoretical tools for the analysis of MaARL. We probe into MaARL from the view of mean-field optimal control. Specifically, our contributions can be summarized as follows:

1. We propose to model MaARL as a mean-field quantitative differential game (Pontryagin, 1985); and thus, its corresponding training process is regarded as how to achieve the optimal control of this game. The mean-field two-sided extremism principle (TSEP) (Guo et al., 2005) is then presented, which relies on the loss function and terminal constraints.

This mean-field TSEP serves as the necessary conditions of the convergence (or equivalently, the optimality) of the mean-field quantitative differential game; when the terminal time is small enough, this mean-field TSEP is also a unique solution, and thus serves as the sufficient conditions of the convergence.

2. The optimal objective function value is characterized by the viscosity solution (E et al., 2019) of a mean-field Hamilton-Jacobi-Issacs (HJI) equation (Guo et al., 2005). We then prove that this viscosity solution is unique. The HJI equation gives a global characterization of adversarial reinforcement learning, while the previously given mean-field TSEP is a local special case.

3. Based on the TSEP, a generalization error bound for MaARL is proved. The bound is of the order $\mathcal{O}(1/\sqrt{N})$, where $N$ is the number of samples. They do not explicitly rely on the dimensions, norms, or other capacity measures of the network parameter, which are usually prohibitively large in deep learning.

To the best of our knowledge, this is the first work on developing theoretical foundations for adversarial reinforcement learning. Our work may inspire novel designs of optimization methods for adversarial reinforcement learning. Moreover, the techniques may be of independent interest in modeling other adversarial learning algorithms, including generative adversarial networks (Goodfellow et al., 2020; Liu and Tuzel, 2016; Mao et al., 2017), and solving partial differential equations (Zang et al., 2020).

## 2    RELATED WORKS.

**Mean-field optimal control.** Since the work of Fornasier and Solombrino (2014) which introduces the concept of the mean-field optimal control and describes it as a rigorous limiting process, various applications of mean-field optimal control in different scenarios were proposed. Fornasier et al. (2019) focus on the role of a government of a large population of interacting agents as a mean-field optimal control problem derived from deterministic finite agent dynamics, Burger et al. (2021) derive a framework to compute optimal controls for problems with states in the space of probability measures, and Albi et al. (2022) studied the problem of mean-field selective optimal control for multi-population dynamics based on transient leadership. In terms of the development of mathematical tools for mean-field optimal control, Bonnet and Frankowska (2022) investigate some of the fine properties of the value function associated with an optimal control problem in the Wasserstein space of probability measures, and Bonnet and Rossi (2021) provide sufficient conditions under which the controlled vector fields solution of optimal control problems formulated on continuity equations are Lipschitz regular in space. See (Bonnet et al., 2022; Zhou and Xu, 2020; Carrillo et al., 2020) for more reference.

**Deep learning theory based on dynamics.** Previous works have been devoted to establish the theoretical foundations of deep learning by the dynamical system viewpoint since E (2017). Based on the PMP and the method of successive approximation (Kantorovitch, 1939), new optimization methods are developed by Li et al. (2018); Li and Hao (2018). Sonoda and Murata (2017) study the continuum limit of training neural networks, and Chang et al. (2018b;a); Haber and Ruthotto (2017) contribute to the design of network architecture based on dynamical systems and differential equations. E et al. (2019) propose to employ mean-field optimal control formulation for explaining deep learning. They prove the mean-field optimality conditions of both the Hamilton-Jacobi-Bellman type and the Pontryagin type (Pontryagin, 1987). Similar results are given by Persio and Garbelli (2021) through associating deep learning with stochastic optimal control (Guo et al., 2005) from the perspective of mean-field games (Lasry and Lions, 2007). These mean-field results reflect the probabilistic nature of deep learning. Compared with above works, this paper models an MaARL algorithm as a mean-field quantitative differential game between two dynamical systems, rather than a single dynamical system.

## 3    PRELIMINARIES

**One-agent reinforcement learning.** One-agent reinforcement learning aims to solve a $K$-step decision problem. Specifically, the agent named $D_z$ starts from state $x^0 = x \in \mathbb{R}^{n_1}$. At step $k$, the agent

are able to take action $a^k$ based on its current state $x^k$ and moves to state $x^{k+1} = x^k + f_k(x^k, a^k)$, where $f_k$ is the transition function. The error of action $a^k$ is penalized by $L_k(x^k, a^k)$ from the environment.

A final penalty $\Phi(x^K, y)$ applies at the last step $K$, where $y$ represents some known prior information, *e.g.*, the engine power of a vehicle in autonomous driving. As a summary, the overall penalties during the agent course is as follows:

$$\mathbb{E}_{(x,y)\sim\mu}\left[\Phi(x^K, y) + \sum_{k=0}^{K-1} L_k(x^k, a^k)\right].$$

In deep reinforcement learning, the action $a^k$ is represented by a deep neural network parameterized by $\theta^k$, i.e., $a^k = a^k(x^k, \theta^k)$, and we can view the agent $D_z$ as a mapping $D_z(x; \hat{\theta}_z = \{\theta^k\}_{k=0}^{K-1})$ from the initial state $x^0 = x$ to the last-step state $x^K$. Also, in practice, the terminal state is usually subject to a constraint represented by function $g(x^K)$, such as a vehicle (the agent) is controlled to reach a certain area (the constraint). In this way, one-agent reinforcement learning can be modelled as a mean-field optimal control problem with trainable parameters $\{\theta^k\}_{k=0}^{K-1}$, as follows,

$$\inf_{\theta} \mathbb{E}_{(x,y)\sim\mu}\left[\Phi(x^K, y) + \sum_{k=0}^{K-1} L_k(x_k, \theta^k)\right]$$

$$s.t. \quad x^{k+1} = x^k + f_k(x^k, \theta^k), x^0 = x \in \mathbb{R}^{n_1}, g(x^K) = 0. \tag{1}$$

**Multi-agent adversarial reinforcement learning.** In this setting, besides the original deep-learning-based agent $D_z(x_z; \hat{\theta}_z = \{\theta_z^k\}_{k=0}^{K-1}) : \mathbb{R}^{n_1} \to \mathbb{R}^{n_1}$ (Eq. 1), an adversarial deep-learning-based agent $D_d(x_d; \hat{\theta}_d = \{\theta_d^k\}_{k=0}^{K-1}) : \mathbb{R}^{n_2} \to \mathbb{R}^{n_2}$ exists, where $\hat{\theta}_z \in \hat{\Theta}_z$ and $\hat{\theta}_d \in \hat{\Theta}_d$ are parameters of $D_z$ and $D_d$ respectively, and $x_z$ and $x_d$ are initial states of $D_z$ and $D_d$ respectively. We use $a_z^k(x_z^k, \theta_z^k)$ and $a_d^k(x_d^k, \theta_d^k)$ to denote the action of the original and the adversarial agents at step $k$, respectively. The penalty at step $k$ now relies on the states and actions both of the original agent $D_z$ and of the adversarial agent $D_d$, represented as $L_k(x_z^k, x_d^k, \theta_z^k, \theta_d^k)$. Similarly, the terminal cost function can be represented as $\Phi(D_z(x_z; \hat{\theta}_z), D_d(x_d; \hat{\theta}_d), y)$.

In MaARL, $\hat{\theta}_z$ is trained to maximize the loss, while $\hat{\theta}_d$ is trained to minimize the loss. We can then formulate the adversarial reinforcement learning problem as

$$\inf_{\hat{\theta}_z \in \hat{\Theta}_z} \sup_{\hat{\theta}_d \in \hat{\Theta}_d} \mathbb{E}_{(x_z, x_d, y)\sim\mu}\left[\Phi(x_z^K, x_d^K, y) + \sum_{k=0}^{K-1} L(x_z^k, x_d^k, \theta_z^k, \theta_d^k)\right] \tag{2}$$

$$s.t. \quad x_z^{k+1} = x_z^k + f_z(x_z^k, \theta_z^k), x_z^0 = x_z, g_z(x_z^K) = 0,$$
$$x_d^{k+1} = x_d^k + f_d(x_d^k, \theta_d^k), k = 0, \dots, K-1, x_d^0 = x_d, g_d(x_d^K) = 0.$$

The goal of MaARL is to find the optimal parameters $\hat{\theta}_z$ and $\hat{\theta}_d$ satisfying Eq. (2), such that two agents reach a Nash equilibrium (Maskin, 1999). In this paper, we consider a currently popular offline setting (Agarwal et al., 2020) of MaARL, where the model is learned on $N$ data points $\{(x_{z_i}, x_{d_i}, y_i)\}_{i=1,\dots,N}$ sampled from the distribution $\mu$.

## 4 MaARL as a Mean-field differential game

Given the current formulation of MaARL (Eq. 2), it is not easy to provide a theoretical analysis due to its discrete iterations. To faciliate analysis, we consider the dynamical systems viewpoint and translate problem (2) into the following continuous form.

$$\inf_{\theta_z} \sup_{\theta_d} J(\theta_z, \theta_d) = \inf_{\theta_z} \sup_{\theta_d} \mathbb{E}_{(x_z, x_d, y)\sim\mu}\left[\Phi(x(t_f), y) + \int_0^{t_f} L(x(t), \theta_z(t), \theta_d(t))dt\right], \tag{3}$$

$$s.t. \quad \frac{dx(t)}{dt} = f(x(t), \theta_z, \theta_d), x(0) = (x_z^T, x_d^T)^T, x(t_f) \in \mathcal{S} := \{x|g(x) = 0\},$$

where $f(x, \theta_z, \theta_d) = (f_z^T(x, \theta_z), f_d^T(x, \theta_d)^T)$, $g(x(t_f)) = (g_z^T(x_z(t_f)), g_d^T(x_d(t_f)))^T$, $x_z(\star) :$ $[0, t_f] \rightarrow \mathbb{R}^{n_1}$, $x_d(\star) : [0, t_f] \rightarrow \mathbb{R}^{n_2}$, $\theta_z(\star) : [0, t_f] \rightarrow \mathbb{R}^{r_1}$, $\theta_d(\star) : [0, t_f] \rightarrow \mathbb{R}^{r_2}$, $g_z(\star) :$ $\mathbb{R}^{n_1} \rightarrow \mathbb{R}^{p_1}$, $g_d(\star) : \mathbb{R}^{n_2} \rightarrow \mathbb{R}^{p_2}$, $\Phi$, $L$ and $f$ are all functions of appropriate input and output dimensions. Thus, $x : [0, t_f] \rightarrow \mathbb{R}^n$, $g : \mathbb{R}^n \rightarrow \mathbb{R}^p$, $n = n_1 + n_2$ and $p = p_1 + p_2$. We define $\mathcal{U}_z$ ($\mathcal{U}_d$) as the set of admissable strategy $\theta_z$ ($\theta_d$) that satisfies the terminal constraint $g_z(x_z(t_f)) = 0$ ($g_d(x_d(t_f)) = 0$).

We note that the above problem (Eq. (3)) is a special case of the mean-field differential games, and name it as the mean-field quantitative differential game. We believe that Eq. (3) is a reasonable modeling of MaARL, since most dynamic systems in MaARL scenarios are naturally described in terms of continuous time because of physical laws (e.g., the trajectory of the vehicle in autonomous driving). Furthermore, as the mean-field quantitative differential game is a special case of the mean-field differential games, methodology from this area can be borrowed and can offer theoretical insight into this problem.

Our goal is to characterize the optimal strategy $(\theta_z^*, \theta_d^*)$ of Eq. (3) and the corresponding optimal trajectory $x^*(t)$ for any $(\theta_z, \theta_d) \in \mathcal{U}_z \times \mathcal{U}_d$ that satisfies

$$J(\theta_z^*, \theta_d) \leq J(\theta_z^*, \theta_d^*) \leq J(\theta_z, \theta_d^*), \tag{4}$$

where Eq. (4) is called the saddle point condition. Furthermore, as Eq. (3) is a characterization of the expected penalty while empirically only the penalty from the sample is available, another goal of us is to characterize the gap between the sampled penalty and the expected penalty.

The rest of the paper is organized as follows: in Section 5, we characterize the optimal solution of Eq. (3) through the mean-field two-sided extremism principle (TSEP). In Section 5, we characterize the optimal objective function value of Eq. (3) through the mean-field HJI function. Finally, in Section 7, we derive the generalization bound between the sampled penalty and the expected penalty.

## 5 MODELING OPTIMAL SOLUTION USING MEAN-FIELD TSEP

In this section, we characterize the optimality of the mean-field quantitative differential game (3) through a two-sided extremism principle (TSEP) with terminal constraints. We prove that satisfying such a TSEP is a necessary condition for being the optimal solution of Eq. (3). With additional mild assumptions, we will show the TSEP is also a sufficient condition.

We first introduce the Hamilton function of Eq. (3) as follows,

$$H(x(t), \theta_z(t), \theta_d(t), \psi(t)) := -L(x(t), \theta_z(t), \theta_d(t)) + \psi^T(t)f(x(t), \theta_z(t), \theta_d(t)),$$

Intuitively, $H : \mathbb{R}^n \times \Theta_z \times \Theta_d \times \mathbb{R}^n \rightarrow \mathbb{R}$ is the total energy of the dynamical system and $\psi \in \mathbb{R}^n$ represents the momentum.

We are now ready to prove derive the necessary condition of being the optimal solution of Eq. (3).

**Theorem 5.1** *Under the assumptions,*

*i) $f$ is bounded and $f, L$ are continuous w.r.t. $\theta_z, \theta_d$;*

*ii) $f, L$ and $\Phi$ are continuously differentiable w.r.t $x$, and the distribution $\mu$ has bounded support.*

*Let $(\theta_z^*, \theta_d^*) \in \mathcal{U}_z \times \mathcal{U}_d$ be the optimal strategy of problem (3), $x^*(t)$ be the corresponding optimal trajectory, then there exists $\psi^* : [0, t_f] \rightarrow \mathbb{R}^n$ and $\xi \in \mathbb{R}^p$ such that for $t \in [0, t_f]$,*

1) $\dot{x}^*(t) = f(x, \theta_z^*, \theta_d^*), \qquad x^*(0) = x_0, \dot{\psi}^*(t) = -\nabla_x H(x^*(t), \theta_z^*(t), \theta_d^*(t), \psi^*(t)),$

$$\psi^*(t_f) = -\nabla_x \Phi(x^*(t_f), y_0) - \xi^T \nabla_x g(x^*(t_f)) \tag{5}$$

2) $\mathbb{E}_{(x_0, y_0) \sim \mu} H(x^*(t), \theta_z^*(t), \theta_d^*(t), \psi^*(t)) = \sup_{\theta_z \in \Theta_z} \inf_{\theta_d \in \Theta_d} \mathbb{E}_{(x_0, y_0) \sim \mu} H(x^*(t), \theta_z, \theta_d, \psi^*(t))$

$$= \inf_{\theta_d \in \Theta_d} \sup_{\theta_z \in \Theta_z} \mathbb{E}_{(x_0, y_0) \sim \mu} H(x^*(t), \theta_z, \theta_d, \psi^*(t)), a.e., \tag{6}$$

*where $f(\cdot)$, $L(\cdot)$ and $\Phi(\cdot)$ are defined in Eq. (3) and $H(\cdot)$ is the Hamilton function.*

Theorem 5.1 introduces the necessary conditions for the convergence of the unique global solution to the mean-field TSEP, relying on the loss function and terminal constraints.

Since the necessary conditions for optimality have been provided by the TSEP, a natural question is to understand when sufficient conditions for optimality can be also provided. This part presents one simple case where it is sufficient, i.e., an optimal solution exists, when does the mean-field TSEP admit a unique solution?

**Theorem 5.2** *Suppose that*

*i) $f$ is bounded, $g$ is continuously differentiable w.r.t $x$ with bounded and Lipschitz partial derivatives, $\mu$ has bounded support in $\mathbb{R}^n \times \mathbb{R}^m$;*

*ii) $f$, $L$ and $\Phi$ are twice continuously differentiable w.r.t $x$, $\theta_z$ and $\theta_d$ with bounded and Lipschitz partial derivatives, and $\partial f/\partial\theta_z\partial\theta_d, \partial L/\partial\theta_z\partial\theta_d \equiv 0$;*

*iii) $H(x, \theta_z, \theta_d, \psi)$ is strongly concave in $\theta_z$, strongly convex in $\theta_d$ and uniform in $x \in \mathbb{R}^n$, $\psi \in \mathbb{R}^n$.*

*Then for sufficiently small $t_f$, if $(\theta_z^1, \theta_d^1)$ and $(\theta_z^2, \theta_d^2)$ are solutions of the mean-field TSEP derived in Theorem 5.1 and are continuously w.r.t time $t$, then $(\theta_z^1, \theta_d^1) = (\theta_z^2, \theta_d^2)$.*

Theorem 5.2 shows that small $t_f$ roughly corresponds to the regime where the reachable set of the forward dynamics is small. Hence, the solution is unique. We then assume the continuity of $\theta_z^1, \theta_d^1, \theta_z^2, \theta_d^2$ with respect to $t$ in Theorem 5.2. In fact, when $\theta_z^1, \theta_d^1, \theta_z^2, \theta_d^2$ are discontinuous on at most a set with zero measure, we can also conclude for a.e. $t \in [0, t_f]$ that $(\theta_z^1(t), \theta_d^1(t)) = (\theta_z^2(t), \theta_d^2(t))$.

# 6 MODELING OPTIMAL OBJECTIVE FUNCTION VALUE VIA MEAN-FIELD HJI EQUATION

In this section, we study mean-field HJI equation from another perspective. This section presents (1) the mean-field HJI equation for MaARL; and (2) the relationship between the HJI equation and the TSEP.

## 6.1 OPTIMAL OBJECTIVE FUNCTION VALUE OBEYS MEAN-FIELD HJI EQUATION

To simplify the notations, we define

$$v^*(t, \mu) := J(\theta_z^*, \theta_d^*, t, \mu), \tag{7}$$

where $J$, $\theta_z^*$, and $\theta_d^*$ are defined in Section 4. One can easily observe that $v^*(t, \mu)$ corresponds to the optimal objective function value with sample distribution $\mu$ and time $t$. we then have following theorem characterizing $v$.

**Theorem 6.1** *Under the assumptions*

*i) $f$, $L$ and $\Phi$ are bounded, and the distribution $\mu \in \mathcal{P}_2(\mathbb{R}^{n+m})$;*

*ii) $f$, $L$ and $\Phi$ are Lipschitz continuous w.r.t $x$ and the Lipschitz constant of $f$ and $L$ are independent of $\theta_z, \theta_d$.*

*Suppose the optimal value function $v^*(t, \mu)$ of Eq. (7) exists, then it is the unique viscosity solution (see the definition in Appendix B) to the following mean-field HJI equation*

$$\partial_t v(t, \mu) + \inf_{\theta_z \in \Theta_z} \sup_{\theta_d \in \Theta_d} \left\{ \int_{\mathbb{R}^{n+m}} [\partial_\mu v(t, \mu)(x, y)]^T [f(x, \theta_z, \theta_d), 0] + L(x, \theta_z, \theta_d) d\mu(x, y) \right\} = 0,$$

$$v(t_f, \mu) = \int_{\mathbb{R}^{n+m}} \Phi(x, y) d\mu(x, y),$$

$$\tag{8}$$

*where $f(\cdot)$, $L(\cdot)$ and $\Phi(\cdot)$ are defined in (3).*

The optimal value function $v^*(t, \mu)$ is the solution to Eq. (8) in Theorem 6.1, revealing the dynamic programming principle, which shows that for any optimal trajectory, starting from any intermediate

state in the trajectory, the remaining trajectory is also optimal. Theorem 6.1 also establishes the uniqueness of the HJI equation with regards to viscosity, and identifies the value function for the mean-field optimal control problem as the unique solution of the HJI equation.

## 6.2 Connection between HJI and TSEP

In Theorem 5.1, we prove the necessary condition of being the optimal solution of Eq. (3) is characterized by the TSEP, while Theorem 6.1 shows the optimal objective value is the unique viscosity solution of the mean-field HJI equation. One may wonder what is the connection between the TSEP and the mean-field HJI equation. In this section, we will show that the TSEP can be understood as a local result compared to the global characterization of the HJI equation. To see this, we will first introduce some basic knowledge on the Wasserstein space and its derivation rules.

### 6.2.1 Derivative in Wasserstein space

Let $D$ represent the Fréchet derivative on Banach spaces. Namely, if $F : U \to V$ is a mapping between two Banach spaces $(U, \|\cdot\|_U)$ and $(V, \|\cdot\|_V)$, then $DF(x) : U \to V$ is a linear operator satisfies

$$\frac{\|F(x+y) - F(x) - DF(x)(y)\|_V}{\|y\|_U} \to 0, \qquad as \; \|y\|_U \to 0. \tag{9}$$

Denote $X \in \mathbb{R}^{n+m}$ as a random variable, we use the shorthand $L^2(\Omega, \mathbb{R}^{n+m})$ for $L^2((\Omega, \mathcal{F}, \mathbb{P}), \mathbb{R}^{n+m})$ to represent the set of $\mathbb{R}^{n+m}$-valued square integrable random variables with respect to a probability measure $\mathbb{P}$. Then we equip this Hilbert space with the norm $\|X\|_{L^2} := (\mathbb{E}\|X\|^2)^{1/2}$. As we assumed in the previous section, $x_0 \in \mathbb{R}^n$, $y_0 \in \mathbb{R}^m$ are random variables and $(x_0, y_0) \sim \mu \in \mathcal{P}_2(\mathbb{R}^{n+m})$, where $\mathcal{P}_2(\mathbb{R}^{n+m})$ denotes the integrable probability measure defined on the Euclidean space $\mathbb{R}^{n+m}$. The space $\mathcal{P}_2(\mathbb{R}^{n+m})$ can be equipped with a metric by 2-Wasserstein distance

$$\mathcal{W}_2(\mu, \nu) := \inf \left\{ \|X - Y\|_{L^2} \Big| X, Y \in L^2(\Omega, \mathbb{R}^{n+m}) \text{ with } \mathbb{P}_X = \mu, \mathbb{P}_Y = \nu \right\}.$$

For $\mu \in \mathcal{P}_2(\mathbb{R}^{n+m})$, define $\|\mu\|_{L^2} := (\int_{\mathbb{R}^{n+m}} \|w\|^2 \mu(dw))^{1/2}$. Now the variable $X \in L^2(\Omega, \mathbb{R}^{n+m})$ if and only if its law $\mathbb{P}_X \in \mathcal{P}_2(\mathbb{R}^{n+m})$. For any function $u : \mathcal{P}_2(\mathbb{R}^{n+m}) \to \mathbb{R}$, we can lift it into its "extension" $U \in L^2(\Omega, \mathbb{R}^{n+m})$ (Cardaliaguet, 2012) by $U(X) = u(\mathbb{P}_X), \forall X \in L^2(\Omega, \mathbb{R}^{n+m})$. In particular, we have that $u$ is $C^1(\mathcal{P}_2(\mathbb{R}^{n+m}))$, if the lifted function $U$ is Fréchet differentiable with continuous derivatives. Since $L^2(\Omega, \mathbb{R}^{n+m})$ can be identified with its dual, if the Fréchet derivative $DU(X)$ exists, by Riesz' theorem, it can be identified with an element of $L^2(\Omega, \mathbb{R}^{n+m})$,

$$DU(X)(Y) = \mathbb{E}[DU(X) \cdot Y], \; \forall Y \in L^2(\Omega, \mathbb{R}^{n+m}).$$

One may check that the law of $DU(X)$ does not depend on $X$ but only on the law of $X$, thus the derivative of $u$ at $\mu = \mathbb{P}_X$ is defined as $DU(X) = \partial_\mu u(\mathbb{P}_X)(X)$, for some function $\partial_\mu u(\mathbb{P}_X) : \mathbb{R}^{n+m} \to \mathbb{R}^{n+m}$.

### 6.3 Derive the characterization

In what follows, we provide the connection between the HJI equation and TSEP. We will show that the TSEP can be understood as a local result compared to the global characterization of the HJI equation. For the value function $v(t, \mu)$ in deduced HJI (equation 8), consider the lifted function $V(t, X)$, where $X = (x, y) \sim \mu$. We define the Hamiltonian for the lifted HJI equation as

$$\mathcal{H}(X, D_X V(t, X)) = \inf_{\theta_z \in \Theta_z} \sup_{\theta_d \in \Theta_d} \mathbb{E}_\mu \big[ D_X V(t, X)^T [f(x, \theta_z, \theta_d), 0] + L(x, \theta_z, \theta_d) \big]. \tag{10}$$

Suppose $\theta_z^\dagger(X, D_X V(t, X))$ and $\theta_d^\dagger(X, D_X V(t, X))$ are the corresponding optimal strategies and define $P = D_X V(t, X)$, we have

$$\mathcal{H}(X, P) = \mathbb{E}_\mu \big[ P^T [f(x, \theta_z^\dagger(X, P), \theta_d^\dagger(X, P)), 0] + L(x, \theta_z^\dagger(X, P), \theta_d^\dagger(X, P)) \big],$$

$$\mathbb{E}_\mu \big[ \nabla_{\theta_z, \theta_d} [f(x, \theta_z^\dagger(X, P), \theta_d^\dagger(X, P)), 0] P + \nabla_{\theta_z, \theta_d} L(x, \theta_z^\dagger(X, P), \theta_d^\dagger(X, P)) \big] = 0, \tag{11}$$

where the last equation follows from the first order optimality condition. Define $X_t = (x_t, y)$, $P_t = D_X V(t, X_t)$, we can apply the characteristic evolution equations (Subbotina, 2006)

$$\dot{X}_t = D_P \mathcal{H}(X_t, P_t), \quad \dot{P}_t = -D_X \mathcal{H}(X_t, P_t). \tag{12}$$

Plugging equation 11 into equation 12, and let $\theta_z^*(t) = \theta_z^\dagger(X_t, P_t), \theta_d^*(t) = \theta_d^\dagger(X_t, P_t)$ and $p_t$ is the first $n$ components of $P_t$, we have

$$\dot{x}_t = f(x_t, \theta_z^*(t), \theta_d^*(t)), \quad \dot{p}_t = -\nabla_x f(x_t, \theta_z^*(t), \theta_d^*(t)) p_t - \nabla_x L(x_t, \theta_z^*(t), \theta_d^*(t)). \tag{13}$$

If we let $\psi = -p$, the first two equalities of equation 5 in Theorem 5.1 is converted to equation 13. The Hamilton equation in TSEP can be regarded as the characteristic equations for the HJI equation originating from $\mu_0$, which justifies the claim that the TSEP constitutes a local condition as compared to the HJI equation.

# 7 GENERALIZATION BOUND

In this section, we establish generalization bounds for MaARL in the offline setting both from the perspective of the global minimum of the loss function and from the perspective of algorithmic stability.

## 7.1 GENERALIZATION BOUND FROM TSEP

We define the loss function of each training sample

$$X_i := (x_{z_i}, x_{d_i}, y_i), \ i = 1, \cdots, N$$

as

$$J^0(\theta_z, \theta_d; X_i) = \Phi(x_z(t_f), x_d(t_f), y_i) + \int_0^{t_f} L(x_z(t), x_d(t), \theta_z(t), \theta_d(t)) dt,$$

where $x_z(0) = x_{z_i}, x_d(0) = x_{d_i}$. Now

$$J(\theta_z, \theta_d) = \mathbb{E}_{X_0 \sim \mu} J^0(\theta_z, \theta_d; X_0), \tag{14}$$

and we define

$$J_N(\theta_z, \theta_d) = \frac{1}{N} \sum_{i=1}^N J^0(\theta_z, \theta_d; X_i). \tag{15}$$

We then estimate the generalization bounds for offline MaARL based on the TSEP. The necessary condition of Hamiltonian for the sampled version is expressed as

$$\frac{1}{N} \sum_{i=1}^N H(x^{\theta_z^N, \theta_d^N, i}(t), \theta_z^N(t), \theta_d^N(t), \psi^{\theta_z^N, \theta_d^N, i}(t))$$

$$= \inf_{\theta_d \in \Theta_d} \sup_{\theta_z \in \Theta_z} \frac{1}{N} \sum_{i=1}^N H(x^{\theta_z^N, \theta_d^N, i}(t), \theta_z, \theta_d, \psi^{\theta_z^N, \theta_d^N, i}(t)),$$

$$a.e., \tag{16}$$

where $t \in [0, t_f]$, $\theta_z^N$ and $\theta_d^N$ are the solution of sampled TSEP. Note that if $\Theta_z$ and $\Theta_d$ are sufficiently large, e.g.

$$\Theta_z = \mathbb{R}^{r_1}, \Theta_d = \mathbb{R}^{r_2},$$

the solution $\theta_z^*, \theta_d^*$ of TSEP satisfies

$$F(\theta_z^*, \theta_d^*)(t) := \mathbb{E}_{\mu_0} \nabla_{\theta_z, \theta_d} H(x_t^{\theta_z^*, \theta_d^*}, \psi_t^{\theta_z^*, \theta_d^*}, \theta_z^*(t), \theta_d^*(t)) = 0, \ a.e. \ t \in [0, t_f],$$

while the solution $\theta_z^N, \theta_d^N$ of sampled TSEP satisfies

$$F_N(\theta_z^N, \theta_d^N)(t) := \frac{1}{N} \sum_{i=1}^N \nabla_{\theta_z, \theta_d} H(x_t^{\theta_z^N, \theta_d^N, i}, \psi_t^{\theta_z^N, \theta_d^N, i}, \theta_z^N(t), \theta_d^N(t)) = 0, \quad a.e. \ t \in [0, t_f]. \tag{17}$$

Now, $F_N$ is a random approximation of $F$ and

$$\mathbb{E}F_N(\theta_z, \theta_d)(t) = F(\theta_z, \theta_d)(t)$$

for all $\theta_z, \theta_d$ and a.e. $t \in [0, t_f]$.

Let $(U, \|\cdot\|_U)$, $(V, \|\cdot\|_V)$ be Banach spaces and $F : U \to V$. We first provide the definition of stability, which is the primary condition that ensures the approximation of $F_N$ to $F$.

**Definition 7.1** *For $\rho > 0$ and $x \in U$, $S_\rho(x) := \{y \in U : \|x - y\|_U < \rho\}$. The mapping $F$ is stable on $S_\rho(x)$ if there exists a constant $K_\rho > 0$ such that,*

$$\|y - z\|_U \leq K_\rho \|F(y) - F(z)\|_V, \quad \forall y, z \in S_\rho(x).$$

Notice that in this case, we are only concerned about whether $\theta_z^*$ and $\theta_d^*$ follow the first-order optimality condition. We define $\theta = (\theta_z^T, \theta_d^T)^T$ and redefine $F(\theta_z^*, \theta_d^*)(\cdot)$ and $F_N(\theta_z^N, \theta_d^N)(\cdot)$ as $F(\theta)(\cdot)$, $F(\theta^N)(\cdot)$, respectively. Then we obtain the following Theorem 7.1, which describes the convergence of the sampled solution to the mean-field solution as the number of samples increases.

**Theorem 7.1** *Assuming that $f$, $L$, and $\Phi$ are bounded and Lipschitz continuous with respect to $x$ and the Lipschitz constants of $f$ and $L$ are independent of $\theta_z, \theta_d$. Let $(\theta_z^*, \theta_d^*)$ be a solution of $F = 0$ (Eq. (17)), which is stable on $S_\rho(((\theta_z^*)^T, (\theta_d^*)^T)^T)$ for some $\rho > 0$. Then there exists positive constants $s_0, C, K_1, K_2, \rho_1 < \rho$ and a random variable $\theta^N := ((\theta_z^N)^T, (\theta_d^N)^T)^T \in S_\rho(((\theta_z^*)^T, (\theta_d^*)^T)^T)$, such that for $s \in (0, s_0]$, the following holds.*

$$\mathbb{P}\big[\|\theta_z^* - \theta_z^N\|_{L^\infty} \geq Cs\big] \leq 4\exp\left\{-\frac{Ns^2}{K_1 + K_2 s}\right\},$$

$$\mathbb{P}\big[\|\theta_d^* - \theta_d^N\|_{L^\infty} \geq Cs\big] \leq 4\exp\left\{-\frac{Ns^2}{K_1 + K_2 s}\right\},$$

$$\mathbb{P}\big[|J(\theta_z^*, \theta_d^*) - J(\theta_z^N, \theta_d^N)| \geq s\big] \leq 4\exp\left\{-\frac{Ns^2}{K_1 + K_2 s}\right\},$$

$$\mathbb{P}\big[F_N(\theta^N) \neq 0\big] \leq 4\exp\left\{-\frac{Ns_0^2}{K_1 + K_2 s_0}\right\}. \tag{18}$$

The loss function (14) is uniformly bounded under the given assumptions, then we can apply the Hoeffding's inequality (Corollary 2 in Pinelis and Sakhanenko (1986)). Using Theorem 6 in E et al. (2019) and rewriting $\theta$ as $(\theta_z^T, \theta_d^T)^T$, this theorem can be proved. Let $s \leq 1$, set the right-hand side of Eq. (18) to be less than $\delta$. Solving for $\epsilon$ immediately yields the following bound.

**Corollary 7.1** *Under the assumptions and notations of Theorem 7.1, for any*

$$0 < \delta \leq \max_{s \in (0, \min\{1, s_0\}]} 4\exp\left\{-\frac{Ns^2}{K_1 + K_2 s}\right\}, \tag{19}$$

*the following inequality holds with probability at least $1 - \delta$.*

$$\|\theta_z^* - \theta_z^N\|_{L^\infty} < C\sqrt{\frac{K_1 + K_2}{N} \log\frac{4}{\delta}},$$

$$\|\theta_d^* - \theta_d^N\|_{L^\infty} < C\sqrt{\frac{K_1 + K_2}{N} \log\frac{4}{\delta}}, \tag{20}$$

$$|J(\theta_z^*, \theta_d^*) - J(\theta_z^N, \theta_d^N)| < \sqrt{\frac{K_1 + K_2}{N} \log\frac{4}{\delta}}.$$

Corollary 7.1 basically shows that the difference between the optimizer over the whole distribution and the optimizer over finite samples is bounded, and has order $\mathcal{O}(1/\sqrt{N})$ with a total of $N$ samples. This bound is independent of the training algorithm.

## 7.2 GENERALIZATION BOUND VIA ALGORITHMIC STABILITY

In the end, we estimate the generalization bounds for offline ARL from the view of algorithmic stability. In the rest of this section, we redefine the integral form in $J$, $J_N$, $J^0$ as the discrete sum form (2) and redefine $r_1, r_2$ as the total dimension of $\theta_z, \theta_d$. We define the generalization error by taking the expectation with respect to the randomized algorithm

$$er(\theta_z, \theta_d) := \mathbb{E}_{\mathcal{A}}\big[J(\theta_z, \theta_d) - J_N(\theta_z, \theta_d)\big].$$

We update $\theta_z$ and $\theta_d$ alternately, i.e. from the initial value $(\theta_{z,0}, \theta_{d,0})$, update $\theta_z$ by $M_{z,1}$ steps to get $(\theta_{z,M_{z,1}}, \theta_{d,0})$, then update $\theta_d$ by $M_{d,1}$ steps to get $(\theta_{z,M_{z,1}}, \theta_{d,M_{d,1}})$. Keep going until the algorithm converges, we can get $(\theta_{z,M_{z,2}}, \theta_{d,M_{d,2}})$, $(\theta_{z,M_{z,3}}, \theta_{d,M_{d,3}}) \cdots (\theta_{z,M_{z,n}}, \theta_{d,M_{d,n}})$.

Consider Stochastic Gradient Langevin Dynamics (SGLD), which is a popular variant of stochastic gradient methods adding isotropic Gaussian noise in each iteration, e.g.

$$\theta_{z,k+1} = \theta_{z,k} - \eta_k \nabla_{\theta_z} J_N(\theta_{z,k}, \theta_{d,0}) + \sqrt{\frac{2\eta_k}{\beta}} \mathcal{N}(0, I_{r_1}).$$

We have the following generalization bound in expectation of random draw of training data.

**Theorem 7.2** *Suppose that $J^0(\theta_z, \theta_d; X)$ is uniformly bounded by C, and*

$$\|\nabla_{\theta_z} J^0(\theta_z, \theta_d; X) - \nabla_{\theta_z} J^0(\theta_z, \theta_d; X')\| \leq L_z,$$
$$\|\nabla_{\theta_d} J^0(\theta_z, \theta_d; X) - \nabla_{\theta_d} J^0(\theta_z, \theta_d; X')\| \leq L_d, \quad \forall X, X',$$

*then we have the following generalization bound*

$$\begin{aligned}
&\mathbb{E}[er(\theta_{z,M_{z,n}}, \theta_{d,M_{d,n}})] \\
&\leq \frac{2}{N} \sum_{i=1}^n \min(k_1, M_{z,i} - M_{z,i-1}) + \frac{\sqrt{\beta} L_z C}{N} \sum_{i=1}^n \left( \sum_{j=k_1+1}^{M_{z,i}-M_{z,i-1}} \eta_j \right)^{1/2} \\
&\quad + \frac{2}{N} \sum_{i=1}^n \min(k_2, M_{d,i} - M_{d,i-1}) + \frac{\sqrt{\beta} L_d C}{N} \sum_{i=1}^n \left( \sum_{j=k_2+1}^{M_{d,i}-M_{d,i-1}} \eta_j \right)^{1/2},
\end{aligned} \tag{21}$$

*where $M_{z,0} = M_{d,0} = 0$, $k_1$ and $k_2$ are chosen to satisfy $\eta_{k_1} \leq \ln 2/\beta L_z^2$, $\eta_{k_2} \leq \ln 2/\beta L_d^2$.*

Theorem 7.2 obtains a bound of $\mathcal{O}(1/N)$, which matches the generalization bounds of stochastic gradient descent ascent (SGDA) for minimax problems in Lei et al. (2021). This bound relies on the aggregated step sizes and does not explicitly depend on the dimensions, norms, or other capacity measures of the parameter, which are usually excessively large in deep learning.

## 8 CONCLUSION

Adversarial reinforcement learning (MaARL) has shown superior performance in solving adversarial games, but the theoretical understanding of MaARL is still premature. This paper studies the convergence and generalization of MaARL under the mean-field optimal control framework. We first model MaARL as a mean-field quantitative differential game problem. We prove the necessary conditions for the convergence of MaARL from two perspectives, two-sided extremism principle (TSEP) and Hamilton-Jacobi-Issacs (HJI) equation. The uniqueness of the solutions to a mean-field TSEP and the HJI equation are also established. Further, we present the connection between a mean-field TSEP and a mean-field HJI equation. In this way, we show that the TSEP is actually a local special case compared to the global characterization of the HJI equation. We also prove two generalization bounds of orders $\mathcal{O}(1/\sqrt{N})$ and $\mathcal{O}(1/N)$ from two aspects, global minimum of the loss function and algorithmic stability, respectively, where $N$ is the number of initial states used in training. Both bounds do not explicitly rely on the dimensions, norms, or other capacity measures of the network parameter, which are usually prohibitively large in deep learning. The bounds illustrate how the algorithmic randomness facilitates the generalization of MaARL. To the best of our knowledge, this is the first theoretical work on the convergence and generalization of MaARL.

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

## A    PROOF OF RESULTS IN SECTION 5

Before proving Theorem 5.2, we write the express in Theorem 5.1 more compactly. For each control process $\theta_z \in L^\infty([0, t_f], \Theta_z)$ and $\theta_d \in L^\infty([0, t_f], \Theta_d)$, we denote by $x^{\theta_z, \theta_d} := \{x_t^{\theta_z, \theta_d} : 0 \le t \le t_f\}$ and $\psi^{\theta_z, \theta_d} := \{\psi_t^{\theta_z, \theta_d} : 0 \le t \le t_f\}$ the solutions of Hamilton's Equation equation 5, i.e.

$$\dot{x}_t^{\theta_z, \theta_d} = f(x_t^{\theta_z(t), \theta_d(t)}, \theta_z, \theta_d), \qquad\qquad x_0^{\theta_z, \theta_d} = x_0,$$
$$\dot{\psi}_t^{\theta_z, \theta_d} = -\nabla_x H(x^{\theta_z, \theta_d}, \theta_z(t), \theta_d(t), \psi_t^{\theta_z, \theta_d}), \qquad \psi_{t_f}^{\theta_z, \theta_d} = -\nabla_x \Phi(x_{t_f}^{\theta_z, \theta_d}, y_0) - \xi \nabla_x g(x_{t_f}^{\theta_z, \theta_d}).$$

We have the following lemma, which provides an estimate of the difference between $x^{\theta_z^1, \theta_d^1}, \psi^{\theta_z^1, \theta_d^1}$ and $x^{\theta_z^2, \theta_d^2}, \psi^{\theta_z^2, \theta_d^2}$.

**Lemma A.1** *Let* $\theta_z^1, \theta_z^2 \in L^\infty([0, t_f], \Theta_z)$ *and* $\theta_d^1, \theta_d^2 \in L^\infty([0, t_f], \Theta_d)$. *Then there exists a constant* $T_0$ *such that for all* $t_f \in [0, T_0)$, *it holds that:*

$$\|x^{\theta_z^1, \theta_d^1} - x^{\theta_z^2, \theta_d^2}\|_{L^\infty} + \|\psi^{\theta_z^1, \theta_d^1} - \psi^{\theta_z^2, \theta_d^2}\|_{L^\infty} \le C(t_f)(\|\theta_z^1 - \theta_z^2\|_{L^\infty} + \|\theta_d^1 - \theta_d^2\|_{L^\infty}),$$

*where* $C(t_f) > 0$ *satisfies* $C(t_f) \to 0$ *as* $t_f \to 0$.

**Proof A.1 (Proof of Lemma A.1)** *Denote $\delta\theta_z := \theta_z^1 - \theta_z^2$, $\delta\theta_d := \theta_d^1 - \theta_d^2$, $\delta x := x^{\theta_z^1,\theta_d^1} - x^{\theta_z^2,\theta_d^2}$ and $\delta\psi := \psi^{\theta_z^1,\theta_d^1} - \psi^{\theta_z^2,\theta_d^2}$. The first two assumptions of Theorem 5.2 leads to*

$$\|\delta x_t\| \leq \int_0^t \|f(x_s^{\theta_z^1,\theta_d^1}, \theta_z^1(s), \theta_d^1(s)) - f(x_s^{\theta_z^2,\theta_d^2}, \theta_z^2(s), \theta_d^2(s))\| ds$$

$$\leq K \int_0^{t_f} \|\delta x_s\| ds + K \int_0^{t_f} \|\delta\theta_z(s)\| ds + K \int_0^{t_f} \|\delta\theta_d(s)\| ds,$$

*and so*

$$\|\delta x\|_{L^\infty} \leq K t_f \|\delta x\|_{L^\infty} + K t_f \|\delta\theta_z\|_{L^\infty} + K t_f \|\delta\theta_d\|_{L^\infty}.$$

*If $t_f \leq T_0 := 1/K$, we have*

$$\|\delta x\|_{L^\infty} \leq \frac{K t_f}{1 - K t_f}(\|\delta\theta_z\|_{L^\infty} + \|\delta\theta_d\|_{L^\infty}). \tag{22}$$

*Similarly,*

$$\|\delta\psi_t\| \leq K\|\delta x_{t_f}\| + K \int_t^{t_f} \|\delta x_s\| + \|\delta\psi_s\| + \|\delta\theta_z(s)\| + \|\delta\theta_d(s)\| ds,$$

$$\|\delta\psi\|_{L^\infty} \leq (K + K t_f)\|\delta x\|_{L^\infty} + K t_f(\|\delta\psi\|_{L^\infty} + \|\delta\theta_z\|_{L^\infty} + \|\delta\theta_d\|_{L^\infty}),$$

*hence*

$$\|\delta\psi\|_{L^\infty} \leq \frac{K(1 + t_f)}{1 - K t_f}\|\delta x\|_{L^\infty} + \frac{K t_f}{1 - K t_f}(\|\delta\theta_z\|_{L^\infty} + \|\delta\theta_d\|_{L^\infty}),$$

*which combined with equation 22 proves the lemma.*

We can now prove Theorem 5.2.

**Proof A.2 (Proof of Theorem 5.2)** *By uniform strong concavity and the second assumption of Theorem 5.2, there exists a $\lambda_0 > 0$ such that*

$$\lambda_0\|\theta_z^1(t) - \theta_z^2(t)\|^2 \leq \big[\mathbb{E}_{\mu_0}\nabla_{\theta_z}H(x_t^{\theta_z^1,\theta_d^1}, \theta_z^2(t), \theta_d^2(t), \psi_t^{\theta_z^1,\theta_d^1})$$
$$- \mathbb{E}_{\mu_0}\nabla_{\theta_z}H(x_t^{\theta_z^1,\theta_d^1}, \theta_z^1(t), \theta_d^1(t), \psi_t^{\theta_z^1,\theta_d^1})\big] \cdot (\theta_z^1(t) - \theta_z^2(t)),$$

$$\lambda_0\|\theta_d^1(t) - \theta_d^2(t)\|^2 \leq \big[\mathbb{E}_{\mu_0}\nabla_{\theta_d}H(x_t^{\theta_z^1,\theta_d^1}, \theta_z^2(t), \theta_d^2(t), \psi_t^{\theta_z^1,\theta_d^1})$$
$$- \mathbb{E}_{\mu_0}\nabla_{\theta_d}H(x_t^{\theta_z^1,\theta_d^1}, \theta_z^1(t), \theta_d^1(t), \psi_t^{\theta_z^1,\theta_d^1})\big] \cdot (\theta_d^2(t) - \theta_d^1(t)).$$

*Note that $\mathbb{E}_{\mu_0}\nabla_{\theta_z,\theta_d}H(x_t^{\theta_z^1,\theta_d^1}, \theta_z^1(t), \theta_d^1(t), \psi_t^{\theta_z^1,\theta_d^1}) = \mathbb{E}_{\mu_0}\nabla_{\theta_z,\theta_d}H(x_t^{\theta_z^2,\theta_d^2}, \theta_z^2(t), \theta_d^2(t), \psi_t^{\theta_z^2,\theta_d^2}) = 0$, $\forall t \in [0, t_f]$ due to the optimality and continuity, then combining the two inequalities above we have*

$$\lambda_0(\|\theta_z^1(t) - \theta_z^2(t)\|^2 + \|\theta_d^1(t) - \theta_d^2(t)\|^2)$$

$$\leq \big[\mathbb{E}_{\mu_0}\nabla_{\theta_z}H(x_t^{\theta_z^1,\theta_d^1}, \theta_z^2(t), \theta_d^2(t), \psi_t^{\theta_z^1,\theta_d^1})$$
$$- \mathbb{E}_{\mu_0}\nabla_{\theta_z}H(x_t^{\theta_z^2,\theta_d^2}, \theta_z^2(t), \theta_d^2(t), \psi_t^{\theta_z^2,\theta_d^2})\big] \cdot (\theta_z^1(t) - \theta_z^2(t))$$

$$+ \big[\mathbb{E}_{\mu_0}\nabla_{\theta_d}H(x_t^{\theta_z^1,\theta_d^1}, \theta_z^2(t), \theta_d^2(t), \psi_t^{\theta_z^1,\theta_d^1})$$
$$- \mathbb{E}_{\mu_0}\nabla_{\theta_d}H(x_t^{\theta_z^2,\theta_d^2}, \theta_z^2(t), \theta_d^2(t), \psi_t^{\theta_z^2,\theta_d^2})\big] \cdot (\theta_d^2(t) - \theta_d^1(t))$$

$$\leq \mathbb{E}_{\mu_0}\|\nabla_{\theta_z}H(x_t^{\theta_z^1,\theta_d^1}, \theta_z^2(t), \theta_d^2(t), \psi_t^{\theta_z^1,\theta_d^1})$$
$$- \nabla_{\theta_z}H(x_t^{\theta_z^2,\theta_d^2}, \theta_z^2(t), \theta_d^2(t), \psi_t^{\theta_z^2,\theta_d^2})\|\|\theta_z^1(t) - \theta_z^2(t)\|$$

$$+ \mathbb{E}_{\mu_0}\|\nabla_{\theta_d}H(x_t^{\theta_z^1,\theta_d^1}, \theta_z^2(t), \theta_d^2(t), \psi_t^{\theta_z^1,\theta_d^1})$$
$$- \nabla_{\theta_d}H(x_t^{\theta_z^2,\theta_d^2}, \theta_z^2(t), \theta_d^2(t), \psi_t^{\theta_z^2,\theta_d^2})\|\|\theta_d^1(t) - \theta_d^2(t)\|$$

$$\leq K(\|\delta x\|_{L^\infty} + \|\delta\psi\|_{L^\infty})(\|\delta\theta_z\|_{L^\infty} + \|\delta\theta_d\|_{L^\infty}).$$

*Combining the above with Lemma A.1, we have*

$$\|\delta\theta_z\|_{L^\infty}^2 + \|\delta\theta_d\|_{L^\infty}^2 \leq \frac{KC(t_f)}{\lambda_0}(\|\delta\theta_z\|_{L^\infty} + \|\delta\theta_d\|_{L^\infty})^2 \leq \frac{2KC(t_f)}{\lambda_0}(\|\delta\theta_z\|_{L^\infty}^2 + \|\delta\theta_d\|_{L^\infty}^2).$$

$C(t_f) \to 0$ *as $t_f \to 0$, by taking $t_f$ sufficiently small, so that $2KC(t_f) < \lambda_0$, which implies $\|\delta\theta_z\|_{L^\infty} = \|\delta\theta_d\|_{L^\infty} = 0$.*

## B    PROOF OF THEOREM 6.1

Now we introduce the definition of viscosity solution. Consider a function $v(t, \mathbb{P}_X) : [0, t_f] \times \mathcal{P}_2(\mathbb{R}^{n+m}) \to \mathbb{R}$, the Hamiltonian $H(X, \partial_{\mathbb{P}_X} v(t, \mathbb{P}_X)(X)) : L^2(\Omega, \mathbb{R}^{n+m}) \times L^2(\Omega, \mathbb{R}^{n+m}) \to \mathbb{R}$ and $\Psi : L^2(\Omega, \mathbb{R}^{n+m}) \to \mathbb{R}$, where $v$ satisfies

$$
\begin{aligned}
\frac{\partial v}{\partial t} + H(X, \partial_{\mathbb{P}_X} v(t, \mathbb{P}_X)(X)) &= 0, && on \ [0, t_f] \times L^2(\Omega, \mathbb{R}^{n+m}), \\
v(t_f, \mathbb{P}_X) &= \Psi(X), && on \ L^2(\Omega, \mathbb{R}^{n+m}).
\end{aligned}
\tag{23}
$$

Then the lifted function $V(t, X) = v(t, \mathbb{P}_X)$ satisfies

$$
\begin{aligned}
\frac{\partial V}{\partial t} + H(X, D_X V(t, X)) &= 0, && on \ [0, t_f] \times L^2(\Omega, \mathbb{R}^{n+m}), \\
V(T, X) &= \Psi(X), && on \ L^2(\Omega, \mathbb{R}^{n+m}).
\end{aligned}
\tag{24}
$$

We say that a bounded, uniformly continuous function $u : [0, t_f] \times \mathcal{P}_2(\mathbb{R}^{n+m}) \to \mathbb{R}$ is a viscosity solution to equation 23 if its lifted function $U : [0, t_f] \times L^2(\Omega, \mathbb{R}^{n+m}) \to \mathbb{R}$ defined by

$$ U(t, X) = u(t, \mathbb{P}_X), $$

is a viscosity solution to the lifted equation equation 24, namely:

i) $U(t_f, X) \leq \Psi(X)$ and for any test function $\gamma \in C^{1,1}([0, t_f] \times L^2(\Omega, \mathbb{R}^{n+m}))$ such that the map $U - \gamma$ has a local maximum at $(t_0, X_0) \in [0, t_f] \times L^2(\Omega, \mathbb{R}^{n+m})$, one has

$$ \partial_t \gamma(t_0, X_0) + H(X_0, D\gamma(t_0, X_0)) \geq 0. $$

ii) $U(t_f, X) \geq \Psi(X)$ and for any test function $\gamma \in C^{1,1}([0, t_f] \times L^2(\Omega, \mathbb{R}^{n+m}))$ such that the map $U - \gamma$ has a local minimum at $(t_0, X_0) \in [0, t_f] \times L^2(\Omega, \mathbb{R}^{n+m})$, one has

$$ \partial_t \gamma(t_0, X_0) + H(X_0, D\gamma(t_0, X_0)) \leq 0. $$

For further details we refer the interested readers to (E et al., 2019).

**Proof B.1 (Proof of Theorem 6.1)** *Suppose $v'(t, \mu)$ is a viscosity solution to equation 8 and $(\theta'_z, \theta'_d)$ is the corresponding optimal strategy.*

*We first fix $\theta'_z$, consider*

$$
\begin{aligned}
\partial_t v_1(t, \mu) + \sup_{\theta_d \in \Theta_d} \left\{ \int_{\mathbb{R}^{n+m}} [\partial_\mu v(t, \mu)(x, y)]^T [f(x, \theta'_z, \theta_d), 0] + L(x, \theta'_z, \theta_d) d\mu(x, y) \right\} &= 0, \\
v_1(t_f, \mu) = \int_{\mathbb{R}^{n+m}} \Phi(x, y) d\mu(x, y). &
\end{aligned}
\tag{25}
$$

*By Theorem 1 and Theorem 2 in (E et al., 2019), $v'(t, \mu)$ is the unique viscosity solution to equation 25 satisfies*

$$
v'(t, \mu) = \sup_{\theta_d \in \mathcal{U}_d} \mathbb{E}_{(x,y) \sim \mu} \left[ \int_t^{t_f} \Phi(x(t_f), y) + L(x(t), \theta'_z(t), \theta_d(t)) dt \right].
\tag{26}
$$

*Then fix $\theta'_d$, similarly we have*

$$
v'(t, \mu) = \inf_{\theta_z \in \mathcal{U}_z} \mathbb{E}_{(x,y) \sim \mu} \left[ \int_t^{t_f} \Phi(x(t_f), y) + L(x(t), \theta_z(t), \theta'_d(t)) dt \right].
\tag{27}
$$

*Now for $(\theta'_z, \theta'_d)$, equation 4 is satisfied, thus $v'(t, \mu) = v^*(t, \mu)$.*

