# OpenReview forum: "Probe Into Multi-agent Adversarial Reinforcement Learning through Mean-Field Optimal Control"
_ICLR.cc/2023/Conference — Submitted to ICLR 2023_

### Official Review · Reviewer_fRuc · 2022-10-24

**Confidence:** 4
**Correctness:** 2
**Technical Novelty And Significance:** 3
**Empirical Novelty And Significance:** Not applicable
**Recommendation:** 3

**Clarity, Quality, Novelty And Reproducibility:**

Although the topic is interesting and the content might be relevant for the mean field control community, I am not convinced that it is relevant for the machine learning community. More explanations on the link between the results and RL would be appreciated. I would also recommend clarifying which results are new and which ones are not (for example Theorem 7.2, but also for the optimality conditions, since it seems that some aforementioned papers are tightly related to the problem studied here).

**Strength And Weaknesses:**

Strength: Applying mean field control to multi-agent problems seem interesting and not well studied.

Weaknesses: The literature review and the link with machine learning should be strengthened, particularly for this type of conference. I provide below more detailed comments:

(1) One part of the literature is not mentioned: multi-player games with mean-field players have already been studied. See for example
- Tembine, H., 2017. Mean-field-type games. AIMS Mathematics, 2(4), pp.706-735
- Cosso, A. and Pham, H., 2019. Zero-sum stochastic differential games of generalized McKean–Vlasov type. Journal de Mathématiques Pures et Appliquées, 129, pp.180-212.
- Carmona, R., Hamidouche, K., Laurière, M. and Tan, Z., 2021. Linear-quadratic zero-sum mean-field type games: Optimality conditions and policy optimization. Journal of Dynamics & Games, 8(4).
Some of these references and the ones cited therein provide optimality conditions. I would recommend clarifying the connection between your results and existing ones.

(2) In the title, and then throughout the text, the authors repeatedly mention that their work contributes to (multi-agent adversarial) reinforcement learning (RL). However, unless I missed something, the work is not directly connected RL. Even the connection with machine learning is not clear to me. There is a gradient-descent type algorithm mentioned in Section 7.2 (but only for one of the two players), for which a bound is provided in Theorem 7.2. But I did not find the proof of this theorem in the appendix. Perhaps it is a direct consequence of an existing result. As a conclusion, I do not clearly see how this paper contributes to the machine learning community. I would appreciate more clarification on this point.

(2) In Theorem 7.2 it seems that some assumptions are not clearly stated. For instance, I imagine that we need to at least ensure that J is differentiable. Please provide a complete set of assumptions.

Typo:
Page 4: “In Section 5” → “In Section 6”


**Summary Of The Paper:**

The paper studies a two-player game from the point of view of mean field control, recently introduced to study optimal control problems in which the goal is to control a distribution. Here, the distribution is the joint distribution of the state at the current time and some information provided at initial time that can be used to compute the terminal total. The authors formulate the problem and then derive optimality conditions. Last, they provide some bounds between the mean-field solution and the solution obtained using an N-sample approximation, as well as some convergence result for a gradient-based algorithm.

**Summary Of The Review:**

Overall, I believe that the comparison with the existing literature (particularly on mean-field type games) should be more detailed, and the main contributions should be more clearly connected to reinforcement learning.

---

### Official Review · Reviewer_HHzm · 2022-10-26

**Confidence:** 4
**Correctness:** 1
**Technical Novelty And Significance:** 2
**Empirical Novelty And Significance:** Not applicable
**Recommendation:** 1

**Clarity, Quality, Novelty And Reproducibility:**

The paper is a very dense theoretical analysis which makes it difficult to understand. I'm not sure if there's much that can be done about that though.

**Strength And Weaknesses:**

The major weakness of this paper is that it appears to ignore the vast literature on theory for multi-agent environments. The authors state "Despite the empirical popularity of MaARL, its theoretical understanding remains blank" and "To the best of our knowledge, this is the first work on developing theoretical foundations for adversarial reinforcement learning." In fact, the whole field of noncooperative game theory is dedicated to a theoretical understanding of multi-agent adversarial environments, and within that field there is a lot of literature on developing a theoretical understanding of reinforcement learning algorithms in multi-agent environments specifically, including convergence bounds.

Consider, for example:
"From Poincaré recurrence to convergence in imperfect information games: Finding equilibrium via regularization" by Perolat et al.
"DREAM: Deep regret minimization with advantage baselines and model-free learning" by Steinberger et al.

Moreover, basing the theoretical analysis on mean-field quantitative differentiable games seems like a strange choice. Why not limit the analysis to a more specific subset of MaARL that fits your assumption?

**Summary Of The Paper:**

This paper proposes to analyze multi-agent adversarial reinforcement learning theoretically through the lens of a mean-field quantitative differentiable game. The authors derive an error bound under their assumptions.

**Summary Of The Review:**

Given the issues with the paper described above, I do not think the paper should be accepted until these issues are addressed.

---

### Official Review · Reviewer_NCJm · 2022-10-29

**Confidence:** 3
**Correctness:** 3
**Technical Novelty And Significance:** 2
**Empirical Novelty And Significance:** 2
**Recommendation:** 5

**Clarity, Quality, Novelty And Reproducibility:**

The clarity of the paper would need improvement to include the agents into account.

**Strength And Weaknesses:**

+: The paper says that "this is the first work on developing theoretical foundations for adversarial reinforcement learning".

-: The paper is not easy to read, or fully get the novelty.

-: Even though the results are around multiple agents, none of the results take the number of agents into account. How are the gaps in terms of scaling users? Note that similar results exist for convergence of MARL to MFC.

-: The paper talks about NE, while we need to have assumptions for uniqueness of such equilibrium for convergence or generalization guarantees. Such discussion will help the paper.

-: Mean field aspects will require information sharing, and that is not realizable in the adversarial setups. This aspect limits the practicality of the setup.

**Summary Of The Paper:**

This paper models Multi-agent adversarial reinforcement learning (MaARL) as a as a mean-field quantitative differential game. A generalization error bound for MaARL is derived in terms of number of samples.

**Summary Of The Review:**

This paper models Multi-agent adversarial reinforcement learning (MaARL) as a as a mean-field quantitative differential game. A generalization error bound for MaARL is derived in terms of number of samples. The key results of the paper need to have error between MaARL and mean field game as the number of agents grow. Further, aspects of uniqueness of equilibrium needs discussion.

---

### Official Review · Reviewer_p9RX · 2022-10-30

**Confidence:** 4
**Correctness:** 2
**Technical Novelty And Significance:** 1
**Empirical Novelty And Significance:** Not applicable
**Recommendation:** 3

**Clarity, Quality, Novelty And Reproducibility:**

The clarity of the paper is good as I find it easy to follow.  The quality and novelty of the paper is questionable as mentioned above.

**Strength And Weaknesses:**

Strength:
- The paper focuses on advancing theoretical results in multi-agent adversarial reinforcement learning which has has limited theory.

Weaknesses:
- My major concerns is the similarity between the results presented in the paper and the one in [1]. Although this paper and [1] deals with a different objective function, once the objective function in MaARL is reformulated as a mean-field optimal control problem, I find the techniques to arrive at the results are pretty similar.
- The structure of the results in the paper is also similar to [1]: theorem 5.1 vs theorem 3 in [1], theorem 5.2 vs theorem 4 in [1], section 6.2 vs section 6.1 in [1].
- I do not find the proof of the results in section 7 in the appendix while I find the results in theorem 7.1 is just the application of theorem 6 in [1].

[1] E. et al. A mean-field optimal control formulation of deep learning

**Summary Of The Paper:**

The paper analyses the solution to the multi-agent adversarial RL (MaARL) problem via mean-field optimal control viewpoint. The authors characterizes the solution using the two-sided extremism principle (TSEP) and HJI equation then provides a connection between them. At the end, the authors provide a generalization bound for MaARL.

**Summary Of The Review:**

My major concerns for this paper is the similarity with the results in E. et al (2018). I find the results presented in the paper is a direct adaptation from E. et al (2018). I hope to see more clarification about the non-triviality and novelty of their results in this paper. Moreover, part of the theoretical results presented in the paper do not have proof.

---

### Decision · Program_Chairs · 2023-01-20

**Decision:**

Reject

**Justification For Why Not Higher Score:**

All reviewers have several major concerns with the paper. No author response is provided.

**Justification For Why Not Lower Score:**

N/A

**Metareview: Summary, Strengths And Weaknesses:**

This paper is a clear reject. The fact author did not provide responses after seeing the reviews is disappointing.